# Validation and Determination of 25(OH) Vitamin D and 3-Epi25(OH)D3 in Breastmilk and Maternal- and Infant Plasma during Breastfeeding

**DOI:** 10.3390/nu12082271

**Published:** 2020-07-29

**Authors:** Jennifer Gjerde, Marian Kjellevold, Lisbeth Dahl, Torill Berg, Annbjørg Bøkevoll, Maria Wik Markhus

**Affiliations:** Institute of Marine Research, P.O box 1870 Nordnes, 5817 Bergen, Norway; Marian.Kjellevold@hi.no (M.K.); Lisbeth.Dahl@hi.no (L.D.); Torill.Berg@hi.no (T.B.); Annbjorg.Bokevoll@hi.no (A.B.); Maria.Wik.Markhus@hi.no (M.W.M.)

**Keywords:** breastmilk, infant, mother, plasma, vitamin D metabolites, 3-Epi25(OH)D3, 25-hvdroxyvitamin

## Abstract

Vitamin D deficiency in pregnant women and their offspring may result in unfavorable health outcomes for both mother and infant. A 25hydroxyvitamin D (25(OH)D) level of at least 75 nmol/L is recommended by the Endocrine Society. Validated, automated sample preparation and liquid chromatography-tandem mass spectrometry (LC-MS/MS) methods were used to determine the vitamin D metabolites status in mother-infant pairs. Detection of 3-Epi25(OH)D3 prevented overestimation of 25(OH)D3 and misclassification of vitamin D status. Sixty-three percent of maternal 25(OH)D plasma levels were less than the recommended level of 25(OH)D at 3 months. Additionally, breastmilk levels of 25(OH)D decreased from 60.1 nmol/L to 50.0 nmol/L between six weeks and three months (*p* < 0.01). Furthermore, there was a positive correlation between mother and infant plasma levels (*p* < 0.01, r = 0.56) at 3 months. Accordingly, 31% of the infants were categorized as vitamin D deficient (25(OH)D < 50 nmol/L) compared to 25% if 3-Epi25(OH)D3 was not distinguished from 25(OH)D3. This study highlights the importance of accurate quantification of 25(OH)D. Monitoring vitamin D metabolites in infant, maternal plasma, and breastmilk may be needed to ensure adequate levels in both mother and infant in the first 6 months of infant life.

## 1. Introduction

Vitamin D plays an important role in bone metabolism. It regulates the calcium and phosphate in the body, making it important for muscle, tooth, and growth development [1]. It may also play an important role in immune system regulation [2,3]. The source of vitamin D in Norwegian diets is in the form of vitamin D2 (ergocalciferol) from plants and vitamin D3 (cholecalciferol) from fish, butter, and eggs [4]. Vitamin D2 and D3 bind to the Vitamin D carrier protein (DBP) before transported to the liver for hydroxylation, producing 25-hydroxyvitamin (25(OH)) D3 and 25(OH)D2. The 25(OH)D2 is derived solely from the diet/supplements and 25(OH)D3 is either derived from the diet/supplements or synthesized in the skin. The C3 epimer forms of vitamin D_3_ have less affinity toward vitamin D protein and even lower affinity for vitamin D receptors compared to 25(OH)D3. The 3-Epi25(OH)D3 binds to vitamin D receptor (VDR) at 2–3% the affinity of 25(OH)D3 [5]. Studies have also shown reduced ability in inducing calcium transport and reduced gene expression in the human colonic carcinoma cell line, Caco-2 [6,7,8]. Determining the concentration of 25(OH)D3 and 3-Epi25(OH)D3 separately is therefore important due to possible difference of effectiveness. However, further studies on the function and source of 3-Epi25(OH)D3 in humans are warranted.

Vitamin D3 has a plasma half-life of 4 to 12 h [9,10] and a circulating half-life of 12 to 24 h [11]. On the other hand, 25(OH)D2 and 25(OH)D3 have longer half-lives, 13 and 15 to 25 days, respectively [12,13,14,15]. The 25(OH)D is further hydroxylated to 1,25-dihydroxyvitamin D (1,25(OH)2D), the most potent physiologically active metabolite with a relatively short half-life of 4 to 6 h [16]. This suggests that measurements of 25(OH)D, 25(OH)D2, and 25(OH)D3 are better indicators of vitamin D status in the blood. Holick et al. have published a set of guidelines for the evaluation of Vitamin D deficiencies [17]. Blood 25(OH)D levels < 50 nmol/L are considered 25(OH)D-deficient, levels between 50 and 75 nmol/L are considered 25(OH)D-insufficient, and >75 nmol/L are considered 25(OH)D-sufficient [17,18,19,20].

Methods which only measure 25(OH)D3 and not 3-Epi25(OH)D3 may cause overestimation of 25(OH)D3 because both analytes would be determined as 25(OH)D3. To obtain accurate measurements of vitamin D levels in mothers and infants, a highly selective and sensitive quantification method for measuring vitamin D metabolites in blood and breastmilk is needed [21]. Thus, in this study, simple, sensitive, and selective liquid chromatography-tandem mass spectrometric (LC-MS/MS) methods were used for determination of 25(OH)D2, 25(OH)D3, and 3-epi-25-hydroxyvitamin D3 (3-Epi25(OH)D3) in plasma, breastmilk, and infant formula. An automated sample preparation involving protein-crash and solid-phase extraction techniques was applied to ensure simple sample treatment, and reduced time and labor.

During pregnancy, the main source of vitamin D for the fetus is the mother, through the umbilical cord. Studies have reported correlation between maternal and infant cord blood 25(OH)D concentrations [22,23,24,25]. Thus, mothers with sufficient 25(OH)D during pregnancy can provide sufficient cord blood concentrations of 25(OH)D crossing the placenta [26,27]. However, vitamin D metabolites levels < 50 nmol/L have been observed in pregnant women and their offspring [28,29,30]. Pregnant women and infants are highly susceptible to vitamin D deficiency which has been associated with adverse health outcomes, such as pre-eclampsia, perinatal complications, postpartum depression, spontaneous abortion, emergency cesarean section delivery, oligohydramnios, polyhydramnios, and gestational diabetes [31,32]. As for the infants, vitamin D deficiency may cause a small-for-gestational age condition, preterm birth, low birth weight, stunting, impaired fetal bone formation, and rickets [31,32,33,34,35]. Accordingly, Norwegian maternal vitamin D < 30 nmol/L has been associated with lower offspring peak bone mass [36]. Personalized vitamin D supplementation during pregnancy and lactation has been suggested [37]. Monitoring vitamin D deficiency in Norwegian mother-infant pairs before and after childbirth is therefore of importance.

The aim of this study was to validate selective and sensitive LC-MS/MS methods for the analysis of vitamin D metabolites with automated sample preparation. These methods were applied to determine the concentration levels of Vitamin D metabolites in mother-infant pairs during the first six months of breastfeeding. Detection and quantification of 3-Epi25(OH)D3 allows evaluation of the impact of 3-Epi25(OH)D3 when assessing vitamin D metabolites status in plasma and breastmilk. The relationships between plasma vitamin D metabolites levels in mother-infant pairs and breastmilk levels were also examined.

## 2. Materials and Methods

### 2.1. Study Population and Design

From January 2016 until February 2017, pregnant women were recruited to participate in a two-armed randomized controlled intervention trial involving cod intake in pregnancy [38]. The study was registered on ClinicalTrials.gov (NCT02610959) in 17 November 2015. All participants were recruited through the womens’ clinic at Haukeland University Hospital in Norway. A total of 137 pregnant women were included in this secondary analysis. Details regarding the main study have been described elsewhere [38].

### 2.2. Biological Samples and Laboratory Analysis

Blood samples were obtained from mother-infant pairs. The mothers were requested to provide a sample of breastmilk at six weeks and three months postpartum. Breastmilk was collected at the beginning, middle, and end of a chosen feed. The three samples were then stored in a freezer until pick-up by study investigators or submission during the third-month follow-up visit. Samples were placed in freezer packs during transport. Upon arrival in the laboratory, samples were stored at –80 °C until analyzed as a pooled sample. Meanwhile, blood sampling was conducted in mothers and infants at three and six months postpartum. Plasma samples from the participants were obtained by collecting blood into BD Vacutainer ^®^ K2E 5.4 mg vials (Franklin Lakes, NJ, USA), centrifuged (1000–1300× *g*, 20 °C, 10 min) within 30 min, and the supernatant was stored at –80 °C until analyzed.

### 2.3. Laboratory Analysis

#### 2.3.1. Chemical and Reagents

The standards 25(OH)D3, 25(OH)D2, 3-Epi25(OH)D3 and internal standard D_6_-25(OH)D3-(26,26,26,27,27,27-D_6_) were obtained from Cerilliant (Round Rock, TX, USA). The internal standards 25(OH)D2-(6,19,19-d_3_) and 3-Epi25(OH)D3-(6,19,19-d_3_), zink sulfate monohydrate, formic acid (analytical grade) and ammonium acetate were purchased from Sigma-Aldrich (St Louis, MO, USA). 

#### 2.3.2. Sample Preparation

All sample preparation and extraction process were performed using robotic a Dual Head MultiPurpose Sampler (MPS XL) equipped with an Anatune CF-100 Centrifuge Option, MicroLiter ITSP Option and Active WashStation [39,40] Using an automated system, samples were prepared by adding 80 μL of internal standards to aliquots of 200-μL samples. Protein-crash, centrifugation, and solid-phase extraction techniques were automated using MPS XL (Anatune, Cambridge, UK). The samples were precipitated with 200 μL zinc sulfate and 500 μL methanol. Samples were then vortexed and centrifuged for 5 min. An aliquot of 500 μL of supernatant was loaded for solid phase extraction and eluted with 40 μL methanol. High purity water (18.2 million ohms, MΩ x cm) was added to the eluted sample prior to LC-MS/MS injection.

#### 2.3.3. LC-MS/MS Procedure

##### Waters Quattro Premier^TM^/XE

LC-MS/MS conditions for the analysis of plasma samples were as follows. For chromatographic separation, acquity ultra performance liquid chromatography (UPLC) (Waters Corporation, Milford, MA, USA) was used. The system was equipped with a degasser, pump, a thermostated acquity sample manager and column oven. Twenty microliters of the sample was injected into the analytical column (Acquity UPLC, HSS PFP 1.8 µm, 2.1 × 100 mm, Waters, Milford, MA, USA). Two millimole per liter ammonium acetate with 0.1% formic acid was used as mobile phase A, while methanol with 0.3% formic acid was used as mobile phase B. The Waters binary solvent manager was programmed as follows: 0–3.0 min, 30% A and 70% B; 3.5–5.0 min, 25% A and 75% B; 5.5–6.0 min, 2% A and 98% B; 6.5–8.0 min, 30% A and 70% B. All gradient steps were linear. For sample detection, a triple-quadrupole mass spectrometry system from Waters Quattro Premier^TM^/XE (Waters Corporation, Milford, MA, USA) was used, equipped with electrospray ion source. MS source parameters are as follows: capillary voltage, 3.0 kV; cone voltage, 15–20 V; source temperature, 120 °C; and cone gas flow rate, 15 L/h. Nitrogen and argon were used as the cone and collision gases, respectively. Parent and fragment ions were detected in multiple-reaction monitoring (MRM) mode and the respective collision energies are listed in Appendix B. Data acquisition for all experiments was carried out with Masslynx V4.1 software (Waters, Milford, USA).

##### Agilent MassHunter

LC-MS/MS conditions for the analysis of plasma, breastmilk, and infant formula samples were as follows: For chromatographic separation, an Agilent 1290 UPLC (Agilent Technologies, Palo Alto, CA, USA) was used. The system was equipped with a degasser, pump, a thermostated autosampler, and column oven. Fifteen microliters of the sample was injected to the analytical column (Acquity UPLC, HSS PFP 1.8 µm, 2.1 × 100 mm, Waters, Milford, MA, USA). Two millimole ammonium acetate with 0.1% formic acid was used as mobile phase A, while methanol with 0.3% formic acid was used as mobile phase B. The Agilent 1290 pump was programmed as follows: 0–3.0 min, 35% A and 65% B; 3.5–4.5 min, 30% A and 70% B; 5.0–6.4 min, 25% A and 75% B; 6.5–8.0 min, 2% A and 98% B; 8.1–9.5 min, 35% A and 65% B. All gradient steps were linear. For sample detection, a triple-quadrupole mass spectrometry system from Agilent 6495B (Agilent Technologies, Santa Clara, CA, USA) was used, equipped with jet stream electronspray ion source. Nitrogen was used as the drying gas, sheath gas, nebulizing gas, and collision gas. The drying and sheath gas temperatures were 200 °C and 300 °C, respectively. The following settings were used: drying gas flow 13 L/min and sheath gas flow: 10 L/min, nebulizer pressure: 40 psi, capillary voltage: 5000 V, and nozzle voltage: 2000 V. Parent and fragment ions were detected in multiple-reaction monitoring (MRM) mode and the respective collision energies are listed in Appendix B. The Agilent MassHunter Workstation software version B.08.00. (Agilent Technologies, Santa Clara, CA, USA) was used to control the LC-MS system, peak integration, quantitation, and calculation.

### 2.4. LC-MS/MS Assay Validation. Linearity, Sensitivity, Precision, Accuracy, and Recovery

Method validation was performed according to bioanalytical method guidelines for biological samples and industry [41,42]. The linearity of the two methods were assessed using six concentration levels of 25(OH)D2, 25(OH)D3, and 3-Epi25(OH)D3, analyzed for ten consecutive days. Sensitivity was determined by calculating the lower limit of detection (LOD) and lower limit of quantification (LOQ). The LOD was determined using the concentration of the lowest diluted sample with signal-to-noise ratio at approximately three. The LOQ was set as the concentration of the lowest standard.

Precision and accuracy were determined by an intra-day and inter-day analysis of in-house quality control samples. Precision was calculated as relative standard deviation (RSD) of experimental concentrations, and the criteria for acceptability was 15% RSD, except for the LOQ where it should not have exceeded 20%. Accuracy was calculated as the comparison between the measured values and nominal sample concentrations. The criteria for acceptability were 15% and 20% (at LOQ) deviation from the nominal values. An in-house quality control plasma sample consisting of 25(OH)D3 at a medium level was used to estimate intra-assay and inter-assay precision and accuracy of 25(OH)D3 for the Waters Quattro PremierTM/XE method. For the Agilent 6495B method, three concentration levels (low, medium, and high) of in-house quality control samples were prepared in plasma, breastmilk, and infant formula. The baseline concentrations of 25(OH)D2, 25(OH)D3, and 3-Epi25(OH)D3 were measured prior to addition of the standard solutions. The 25(OH)D2, 25(OH)D3, and 3-Epi25(OH)D3 were then assayed for intra-assay precision and accuracy for ten consecutive days. The inter-assay precision was evaluated for 25(OH)D2, 25(OH)D3, and 3-Epi25(OH)D3. Accuracy was determined using commercial quality control (QC) plasma samples; standard reference materials (SRM) 972a-C level 1, SRM 972a-C level 2, SRM 972a-C level 3, and SRM 972a-C level 4 Vitamin D Metabolites in frozen human serum. SRM 1950 metabolites in frozen human plasma were also used. In addition, QC serum samples were also included (SRM 972 level 2, SRM 1950, and SRM 2972). The recovery of the analytes was studied by spiking the samples (plasma, breastmilk, and infant formula) with standard solutions at three levels. Recovery was evaluated by comparing the baseline (unspiked) and spiked samples at three levels (low, medium, and high). Recovery was then calculated by comparing the measured concentration of the prepared samples with the nominal value (baseline and spiked concentration) representing 100% recovery.

### 2.5. Data Analysis and Statistical Analyses

LC-MS/MS data were analyzed by Agilent MassHunter 8.0 Quan Browser (Agilent Technologies, Santa Clara, CA, USA). The levels of vitamin D metabolites are presented as mean and standard deviation (SD). To examine differences between status and categories paired Student’s t-test was used. Pearson’s correlation coefficients were used to assess associations between continuous variables. A 2-sided *p*-value of <0.05 was considered significant. All statistical analyses were performed using IBM SPSS Statistics 26 (IBM Corp., Armonk, NY, USA).

## 3. Results

### 3.1. Evaluation of the LC-MS/MS Assay

The LC MS/MS methods for the determination of 25(OH)D2, 25(OH)D3, and 3-Epi25(OH)D3 using Waters Quattro Premier^TM^/XE, and 25(OH)D2, 25(OH)D3, and 3-Epi25(OH)D3 using Agilent 6495B were developed and validated. The base peak ions and fragments of 25(OH)D2, 25(OH)D3, and 3-Epi25(OH)D3 ([M + H]^+^) are shown in Appendix B. The Waters Quattro Premier^TM^/XE is equipped with an electronspray ion source and Agilent 6495B is equipped with a jet stream electronspray ion source. The selected reaction monitoring was based on the mass-to-charge ratio (*m*/*z*). The Agilent 6495B MRM chromatograms of the analytes are shown in the Appendix A. It provided separation between metabolites, with 25(OH)D3, 25(OH)D2, and 3-Epi25(OH)D3 eluting at 6.95, 7.18, and 7.19 min, respectively. The same was observed using Waters Quattro Premier^TM^/XE with the mass-to-charge ratio (*m*/*z*) and fragmentations distinguished for Waters Quattro Premier^TM^/XE (data not shown).

The Waters Quattro Premier^TM^/XE method was linear for 25(OH)D3, 25(OH)D2, and 3-Epi25(OH)D3 as shown in Table 1, and the best fits were indicated by a correlation coefficient (r) of 0.99 for 25(OH)D3, 25(OH)D2, and 3-Epi25(OH)D3. The Agilent 6495B method was linear for 25(OH)D3, 25(OH)D2, and 3-Epi25(OH)D3 with correlation coefficients ≥ 0.999. The standard curves are considered acceptable when r is >0.99. The limit of detection (LOD) and limit of quantification (LOQ) were determined to evaluate sensitivity of the method and are shown in Table 1a for Quattro Premier^TM^/XE and Table 1b for Agilent 6495B.

The intra-assay and inter-assay precision and accuracy are summarized in Table 2 and Table 3. Relative standard deviation (RSD) values for quality control (QC) plasma, breastmilk, and infant formula samples measured with Waters Quattro Premier^TM^/XE and with Agilent 6495B were below 20% for 25(OH)D2, 25(OH)D3, and 3-Epi25(OH)D3, respectively. For low concentration levels, RSD value for breastmilk was 30% for 3-Epi25(OH)D3. The methods showed accuracy within 20%, as shown in Appendix C. The recoveries of the analytes obtained with the use of automated solid phase extraction and filtration autosampler, Gerstel multi-purpose sampler, ranged from 88% to 120%, 72% to 103% and 81% to 129% for plasma, breastmilk, and infant formula, respectively as shown in Appendix D.

### 3.2. Vitamin D Levels at Each Time-Point for Maternal and Infant Plasma Samples

Table 4 shows the number of samples collected. The samples were divided into 3 diagnostic categories according to 25(OH)D status: deficient (25(OH)D < 50 nmol/L), insufficient (50 nmol/L < 25(OH)D < 75 nmol/L), and sufficient (25(OH)D ≥ 75 nmol/L). The 3-Epi25(OH)D3 was not included in the calculations. Eighteen percent and 30.6% of the measured breastmilk samples had 25(OH)D levels lower than 50 nmol/L, 39.3% and 40% had levels between 50 and 75 nmol/L, and 42.7% and 29.4% had levels equal to or more than 75 nmol/L at 6 weeks and 3 months, respectively. Prevalence of 25(OH)D deficiency was determined in mother-infant pairs (Table 4). At 3 months, 63% of maternal 25(OH)D plasma levels were less than 75 nmol/L 25(OH)D. Accordingly, 31% of the infants were categorized as vitamin D deficient (25(OH)D < 50 nmol/L).

### 3.3. Maternal and Infant Plasma Vitamin D Metabolites Concentration at 3 and 6 Months

The 25(OH)D2, 25(OH)D3, and 3-Epi25(OH)D3 plasma levels of the mothers and their infants, and the breastmilk levels of vitamin D metabolites at 6 weeks and 3 months are presented in Table 5. At 3 months, the concentration of 3-Epi25(OH)D3 contributed 11% and 24% of the total 25(OH)D3 in infants’ plasma and breastmilk, respectively.

### 3.4. Correlation between Vitamin D Metabolites and Mother and Infant Levels of Vitamin D Metabolites

At 6 weeks, a positive association between breastmilk concentration of 25(OH)D3 and 3-Epi25(OH)D3 (*p* < 0.01; r = 0.711, Figure 1a) was observed. The same was seen at 3 months (*p* < 0.01; r = 0.805, Figure 1b). There was also a positive association observed between the infant’s plasma levels of 25(OH)D3 and 3-Epi25(OH)D3 (*p* < 0.01, r = 0.678) at 3 months. A positive association between mother and infant plasma levels of 25(OH)D3 was observed (*p* < 0.01; r = 0.555, Figure 2).

## 4. Discussion

In this study, a robotic autosampler was used to automate sample preparation prior to LC-MS/MS analysis which included protein-crash, solid-phase extraction techniques and dilution. This ensured reproducibility, simple sample treatment, and reduced time and labor and costs. Two simple, sensitive, and selective LC-MS/MS methods were then used to separate and detect 25(OH)D2, 25(OH)D3, and 3-Epi25(OH)D3. The concentrations of 25(OH)D2, 25(OH)D3, and 3-Epi25(OH)D3 in plasma were measured with Waters Quattro PremierTM/XE. Breastmilk samples were measured with Agilent 6495B which resulted in an improved detection limit. Both LC-MS/MS methods separated 3-Epi25(OH)D3 from 25(OH)D3 for accurate detection and measurement of 25(OH)D3. Our validation showed a concentration-response relationship fitted with a simple regression model. The accuracy and inter-precision were within the acceptable criteria of RSD at 15% for 25(OH)D2, 25(OH)D3, and 3-Epi25(OH)D3 in plasma, breastmilk, and infant formula [42] with an exception of the measured inter-precision RSD for 25(OH)D2 in infant formula (17%.) At LOQ levels, the 25(OH)D2, 25(OH)D3, and 3-Epi25(OH)D3 were also within the acceptable criteria of 20% RSD. The intra-precision was within acceptable criteria of 15% RSD. This also applies at LOQ levels with an exception for 3-Epi25(OH)D3 in breastmilk with an RSD of 30%. These two LC-MS/MS methods were used to determine the vitamin D status in Norwegian mother-infant pairs.

The LC-MS/MS methods presented in this study distinguished between 25(OH)D3 and 3-Epi25(OH)D3. At 3 months, the C-3 epimer contributed 11% of the total 25(OH)D in infants’ plasma. Thus, if 3-Epi25(OH)D3 was not distinguished from 25(OH)D3, only 25% of the infants are categorized as vitamin D deficient instead of 31%. The observed positive correlation between infants’ 25(OH)D3 and 3-Epi25(OH)D3 at 3 months suggests possible overestimation of 25(OH)D3 and more likely, misclassification of vitamin D in studies using methods that only measure 25(OH)D3 [43]. In breastmilk, at 3 and 6 months, the 3-Epi25(OH)D3 contributed 24% and 27% of the total 25(OH)D, respectively. In addition, positive associations were also observed between 25(OH)D3 and 3-Epi25(OH)D3 in breastmilk at 6 weeks and 3 months. Accordingly, studies have shown positive correlation between 25(OH)D3 and 3-Epi25(OH)D3 in both maternal and cord blood [43,44]. This may imply maternal cord blood and breastmilk as a possible source of 3-Epi25(OH)D3 in infants. Studies on epimers function and source are warranted.

Infants are highly dependent on their mother’s cord 25(OH)D concentrations, which is usually 60–80% of maternal values at delivery [45,46]. A survey of 25(OH)D levels in pregnant Black South Africans gave a mean of 57.0 nmol/L, while 41.9 nmol/L was found in the cord blood [47]. On the other hand, the measured concentrations of 25(OH)D in light-skinned pregnant women at northern latitudes was 71.4 nmol/L, while 39.2 nmol/L was recorded for cord blood [48]. Another study observed that in pregnant women, the mean 25(OH)D concentrations were 41 nmol/L and 50.7 nmol/L during the first trimester and in the umbilical cord, respectively [49]. Meanwhile, vitamin D deficiency appears to be common in mothers and their infants in New Zealand, with mean cord blood 25(OH)D value of 41 nmol/L [50]. In general, studies suggest that low levels of 25(OH)D in mothers are likely to be reflected in their infants. In this study, maternal plasma levels, at 3 and 6 months, 13% and 14% of mothers were categorized as deficient (25(OH)D < 50 nmol/L), 51% and 48% as insufficient (50 nmol/L < 25(OH)D < 75 nmol/L), and 37% and 38% as sufficient (25(OH)D ≥ 75 nmol/L), respectively. However, 31% and 29% of the infants had deficient and insufficient 25(OH)D status at 3 months, respectively. Vitamin D body stored in infants can decline by 50% over less than a month, hence without another source of vitamin D, vitamin D deficiency can develop [51]. This may explain the higher prevalence of 25(OH)D deficiency in infants compared with mothers.

A positive association between maternal 25(OH)D plasma levels and infant 25(OH)D plasma levels at 3 months was observed. The maternal and infant 25(OH)D3 plasma concentrations were 69.6 (±16.3) nmol/L and 64.9 (±29.1) nmol/L, respectively. Hence, measured infant plasma level of 25(OH)D at 3 months may reflect the 25(OH)D transferred from mother to infant during pregnancy and start of vitamin D supplement. There was a rise of infant plasma 25(OH)D concentration observed from 3 months to 6 months. Accordingly, a positive correlation between maternal and infant plasma was no longer observed suggesting an increase in 25(OH)D level as a possible result of vitamin D supplementation and start of fortified complementary feeding. In Norway, vitamin D supplements, such as vitamin D drops or cod liver oil, are recommended for all infants from the age of four weeks. A Norwegian population study showed that 92% of the children at age 9-16 months were given vitamin D containing supplements [30]. Here, results showed decreased prevalence of 25(OH)D deficiency in infants, from 3 months (31%) to 6 months (8%). Even with the decrease in vitamin D deficiency prevalence at 6 months, vitamin D deficiency originating in the intrauterine period or immediately after birth to 3 months may still result in adverse effects on skeletal development [49,52]. Correspondingly, in vitamin D-deficient infants, rickets usually develops between the age of 6 months and 2 years [53]. Interestingly, the prevalence of infants categorized as 25(OH)D sufficient at 3 and 6 months had a minor change: 40% and 44%, respectively.

The WHO’s recommendation of exclusive breastfeeding for 6 months may further lead to vitamin D deficiency among infants [54,55]. Although breastmilk is rich in essential nutrients for the earliest life stage, it contains about three-times-less vitamin D than the maternal circulating concentration [56]. A study showed that during the third trimester of pregnancy, maternal 25(OH)D was 60 nmol/L, and the breastmilk level at delivery was 1.26 nmol/L [57]. As for non-Vitamin-D-supplemented Norwegian mothers and infants, a study reported no change in mother’s plasma levels of 25(OH)D (58 to 42 nmol/L) between 4 days and 6 weeks after birth. However, infant’s plasma levels dropped from 26 to 15 nmol/L, thus, suggesting that Norwegian mothers have insufficient 25(OH)D, and therefore that breastmilk is inadequate as a lone source of vitamin D, and supplementation of 10 µg per day is recommended [18,58,59].

There was no observed change in maternal 25(OH)D plasma levels between 3 and 6 months. The prevalence of maternal vitamin D deficiency at 3 and 6 months was almost the same. Likewise, another study showed that the measured 25(OH)D3 of 67 nmol/L did not change between 2 weeks and 12 months postpartum [60]. Thus, maternal vitamin status was maintained through 6 months of breastfeeding and possibly just enough for their own needs. These results do not support the theory that maternal vitamin D status decreases due to a transfer of vitamin D from the mother to infant through breastmilk [61]. Accordingly, breastmilk 25(OH)D2 remained unchanged between 6 weeks and 3 months. In contrast, breastmilk concentration of 25(OH)D3 decreased from 6 weeks to 3 months. Thus, decreased level of 25(OH)D3 in breastmilk suggests that transfer of 25(OH)D3 to breastmilk production was not prioritized. Studies have shown that vitamin D is readily transferred into breastmilk, while 25(OH)D is transferred very poorly, and 1,25 (OH)2D is not transferred at all [62]. This may explain the lack of association between maternal plasma and breastmilk 25(OH)D3 concentration. Thus, 25(OH)D in the blood may not be a good reflection of the available amount of vitamin D that can be transferred to infants through breastfeeding. Accordingly, studies have also reported that UV exposure and increased maternal vitamin D supplementation showed minimal changes in 25(OH)D concentrations but a profound increase in breastmilk concentration of vitamin D was observed [56,63]. Thus, measurement of vitamin D3 and vitamin D2 in breastmilk and plasma samples in this study population is warranted.

In this study, an automated robust sample preparation was used. The methods distinguished between 25(OH)D3 and 3 epimer of 25(OH)D3, 3-Epi25(OH)D3 preventing overestimation of 25(OH)D3 and misclassification of vitamin D status. However, a limitation in this study is that two different LC-MS/MS methods were used, resulting in different validation parameters and LOD. Anyhow, the challenge with the overestimation of 25(OH)D3 has been addressed and managed resulting in a reliable quantification of 25(OH)D. Here, we provided novel results of 25(OH)D status in Norwegian mother-infant pairs during the first 6 months of breast feeding. A possible source of 3-epi-25(OH)2D3 was also discussed. Prevalence of vitamin D deficiency (25(OH)D < 50 nmol/L) among Norwegian mother-infant pairs was determined. Surprisingly, a prevalence of 31% 25(OH)D deficiency was observed among infants at three months. Studies suggest the importance of vitamin D supplementation during pregnancy, and the breastfeeding period to ensure the achievement of the recommended 25(OH)D level of at least 75 nmol/L in mothers and their infants [17,20,34,64,65]. Vitamin D supplementation in pregnant women seems to reduce the risk of pre-eclampsia, gestational diabetes, low birthweight, and severe postpartum hemorrhage [66]. Wide ranges of vitamin D doses, from 400 to 2000 IU have been recommended [17,67,68,69]. However, some doses may not result in optimal 25(OH)D levels during pregnancy. This may be due to interindividual variability in vitamin D metabolism and other factors [70,71]. Thus, establishing a personalized dosing regimen has been suggested [37].

## 5. Conclusions

In summary, two simple, sensitive, and selective LC-MS/MS methods enable reliable quantification of 25(OH)D2, 25(OH)D3, and 3-Epi25(OH)D3. The automated sample preparation makes it suitable for routine laboratory analysis of plasma, breastmilk, and infant formula. This study demonstrated the importance of separating 3-Epi25(OH)D3 from 25(OH)D3 to prevent overestimation of 25(OH)D3 and misclassification of vitamin D status. Accordingly, at 3 months, 13% and 31% of the mothers and infants were categorized as vitamin D deficient (25(OH)D < 50 nmol/L), respectively. Thus, we suggest monitoring vitamin D metabolites in infant, maternal plasma, and breastmilk to ensure adequate levels in both mother and infant in the first 6 months of infant life. Further studies on personalized supplementation with vitamin D during pregnancy and breastfeeding are warranted.

## Figures and Tables

**Figure 1 nutrients-12-02271-f001:**
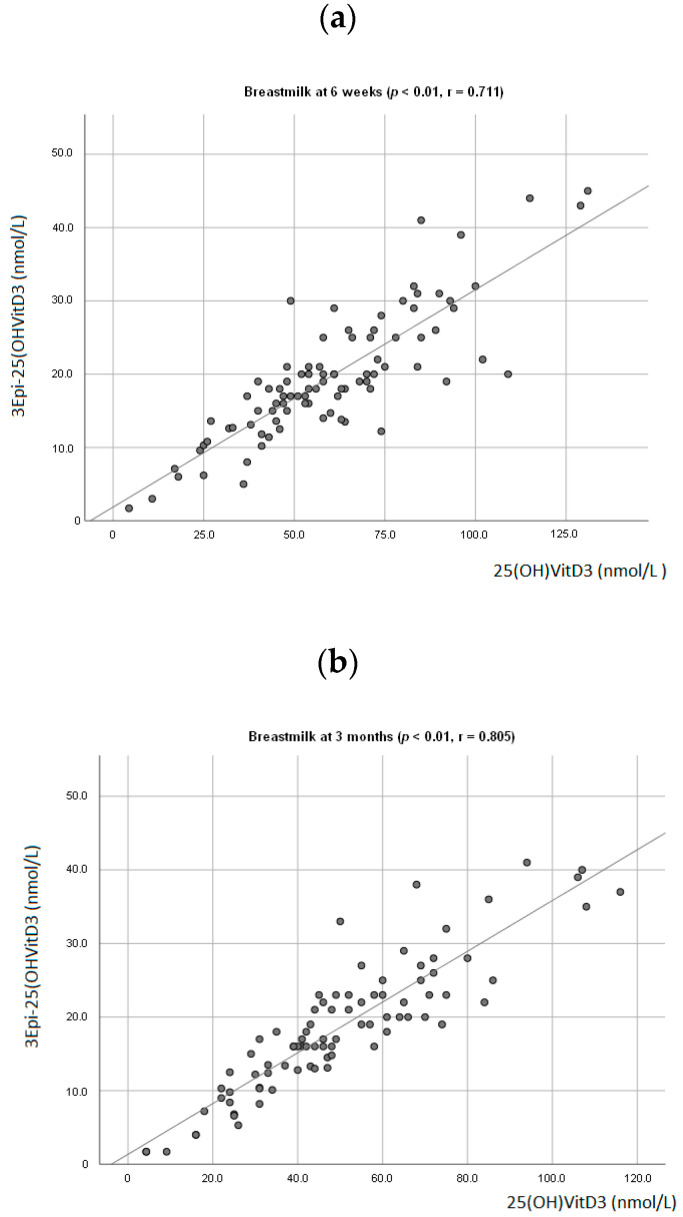
Association between mother’s breastmilk concentration level of 25(OH)D3 and 3-Epi25(OH)D3 at 6 weeks postpartum (**a**) and 3 months postpartum (**b**).

**Figure 2 nutrients-12-02271-f002:**
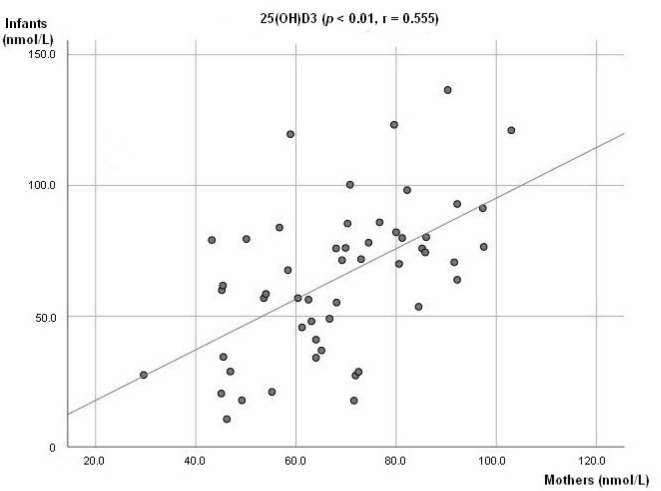
Association between mother’s and infant concentration level of 25(OH)D3.

**Table 1 nutrients-12-02271-t001:** Linearity, limit of quantification, and limit of detection of vitamin D metabolites measured by (a) Waters Quattro Premier^TM^/XE and (b) Agilent 6495B.

	Linearity	r	LOQ	LOD
**a. Waters Quattro Premier^TM^/XE**	**nmol/L**		**nmol/L**	**nmol/L**
25(OH)D2	7.6–242.3	0.9969	7.6	2.3
25(OH)D3	7.8–249.6	0.9985	7.8	2.34
3-Epi25(OH)D3	7.8–249.6	0.9986	7.8	2.3
**b. Agilent 6495B**				
25(OH)D2	7.0–335	0.9999	7.0	2.1
25(OH)D3	4.3–270	0.9998	4.3	1.3
3-Epi25(OH)D3	1.7–137	0.9999	1.7	0.5

Limit of quantification (LOQ), limit of detection (LOD).

**Table 2 nutrients-12-02271-t002:** Intra-day (*n*) of vitamin D metabolites in plasma, breastmilk, and infant formula samples measured by (a) Waters Quattro Premier^TM^/XE and (b) Agilent 6495B.

Intra-Day Precision	Vitamin D Metabolites		
**a. Waters Quattro Premier^TM^/XE**			
**Plasma (*n* = 6)**	25(OH)D2	25(OH)D3	3-Epi25(OH)D3
Medium1			
Average (nmol/L)	-	55.7 ± 4.8	-
RSD (%)		8	
**b. Agilent 6495B**			
**Plasma (*n* = 10)**	25(OH)D2	25(OH)D3	3-Epi25(OH)D3
Low2			
Average (nmol/L)	31.5 ± 4.4	46.1 ± 2.8	34.6 ± 2.7
RSD (%)	14	6	8
Medium2			
Average (nmol/L)	76.2 ± 7.5	111.0 ± 3.9	49.3 ± 3.5
RSD (%)	10	3	7
High2			
Average (nmol/L)	120.3 ± 8.9	177.9 ± 14.6	105.6 ± 5.6
RSD (%)	7	8	5
**Breastmilk (*n* = 10)**	25(OH)D2	25(OH)D3	3-Epi25(OH)D3
Low3			
Average (nmol/L)	<LOQ	11.7 ± 1.9	6.8 ± 1.0
RSD (%)		16	14
Medium3			
Average (nmol/L)	47.8 ± 3.0	71.6 ± 2.5	51.7 ± 2.3
RSD (%)	6	4	4
High3			
Average (nmol/L)	103.1 ± 4.4	126.8 ± 9.4	96.7 ± 4.6
RSD (%)	4	7	5
**Infant formula (*n* = 10)**	25(OH)D2	25(OH)D3	3-Epi25(OH)D3
Low4			
Average (nmol/L)	<7.6	<7.8	<7.8
RSD (%)			
Medium4			
Average (nmol/L)	18.1 ± 3.0	75.3 ± 6.6	31.1 ± 1.7
RSD (%)	17	9	5
High4			
Average (nmol/L)	88.5 ± 8.0	119.2 ± 6.1	109 ± 3.3
RSD (%)	9	5	3

Relative standard deviation (RSD %).

**Table 3 nutrients-12-02271-t003:** Inter-day precision (*n*) of vitamin D metabolites in plasma, breastmilk, and infant formula samples measured by (**a**) Waters Quattro Premier^TM^/XE and (**b**) Agilent 6495B.

	Analyte		
**a. Waters Quattro Premier^TM^/XE**			
**Plasma (*n* = 10)**	25(OH)D2	25(OH)D3	3-Epi25(OH)D3
Medium1			
Average (nmol/L)	-	55.5 ± 5.2	-
RSD (%)		9	
**b. Agilent 6495B**			
**Plasma (*n* = 10)**	25(OH)D2	25(OH)D3	3-Epi25(OH)D3
Low2			
Average (nmol/L)	30.2 ± 2.7	42.9 ± 3.4	34.5 ± 2.2
RSD (%)	9	8	6
Medium2			
Average (nmol/L)	71.2 ± 4.2	102.6 ± 5.6	48.2 ± 3.6
RSD (%)	6	5	7
High2			
Average (nmol/L)	115.3 ± 8.9	162.7 ± 9.0	99.9 ± 5.6
RSD (%)	8	6	6
**Breastmilk (*n* = 10)**	25(OH)D2	25(OH)D3	3-Epi25(OH)D3
Low3			
Average (nmol/L)	<7	12.0 ± 2	5.8 ± 1.7
RSD (%)		17	30
Medium3			
Average (nmol/L)	39.9 ± 2.7	65.0 ± 5.2	45.7 ± 2.6
RSD (%)	7	8	6
High3			
Average (nmol/L)	97.9 ± 4.6	114.8 ± 8	92.0 ± 9
RSD (%)	5	7	10
**Infant formula (*n* = 10)**	25(OH)D2	25(OH)D3	3-Epi25(OH)D3
Low4			
Average (nmol/L)	<7	<4.3	<1.7
SD			
RSD (%)			
Medium4			
Average (nmol/L)	21.4 ± 2.3	66.6 ± 2.7	32.2 ± 1.8
RSD (%)	11	4	6
High4			
Average (nmol/L)	93.2 ± 8.7	112.1 ± 4.1	107.1 ± 3.6
RSD (%)	9	4	3

**Table 4 nutrients-12-02271-t004:** Distribution of vitamin D status at each time-point (6 weeks, 3, and 6 months) for maternal and infant samples.

		Weeks	Months	
**Mothers**		6	3	6
Breastmilk	*n* ^1^	89 (%)	85 (%)	
Deficient	<50 nmol/L	16 (18.0)	26 (30.6)	
Insufficient	50–74 nmol/L	35 (39.3)	34 (40.0)	
Sufficient	≥75 nmol/L	38 (42.7)	25 (29.4)	
Plasma	*n* ^1^		87 (%)	84 (%)
Deficient	<50 nmol/L		11 (12.6)	12 (14.3)
Insufficient	50–74 nmol/L		44 (50.6)	40 (47.6)
Sufficient	≥75 nmol/L		32 (36.8)	32 (38.1)
**Infants**				
Plasma	Total		52 (%)	48 (%)
Deficient	<50 nmol/L		16 (30.8)	4 (8.3)
Insufficient	50–74 nmol/L		15 (28.8)	23 (47.9)
Sufficient	≥75 nmol/L		21 (40.4)	21 (43.8)

^1^ number of samples (*n*).

**Table 5 nutrients-12-02271-t005:** Concentration levels of 25(OH)D2, 25(OH)D3, and 3-Epi25(OH)D3 at 6 weeks, 3, and 6 months among mother and infant. Paired samples tests.

		Weeks	Months		*p* Value ^1^
		6	3	6	
**Mothers**		nmol/L	nmol/L		
Breastmilk	25(OH)D2	12.8 ± 6.7	12.7 ± 7.1	-	0.728
	25(OH)D3	60.1 ± 24.8	50.0 ± 23.4	-	0.001
	3-Epi25(OH)D3	19.7 ± 8.7	18.6 ± 9.0	-	0.327
			nmol/L	nmol/L	
Plasma	25(OH)D2		<7.6 ^2^	<7.6	
	25(OH)D3		69.6 ± 16.3	69.6 ± 19.0	0.526
	3-Epi25(OH)D3		<7.8	<7.8	
**Infants**					
Plasma	25(OH)D2		<7.6	<7.6	
	25(OH)D3		64.6 ± 29.1	83.1 ± 27	0.001
	3-Epi25(OH)D3		7.8 ± 4.8	<7.6	

^1^ Paired samples test. ^2^ The number of breastmilk samples below limit of quantification (LOQ) were 31 and 32 for 25(OH)D2, 0 and 2 for 25(OH)D3, and 1 and 3 for 3-Epi25(OH)D3 at 6 weeks and 3 months, respectively. The number of infant plasma samples below LOQ were 19 and 20 for 3-Epi25(OH)D3 at 3 and 6 months, respectively.

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
