# Peer review of "Validation and Determination of 25(OH) Vitamin D and 3-Epi25(OH)D3 in Breastmilk and Maternal- and Infant Plasma during Breastfeeding"

_nutrients, 2020, doi:10.3390/nu12082271_

Round 1

Reviewer 1 Report

The authors of the study analyze, in pairs, mothers (in plasma and breast milk) and children, the state of Vitamin D status and its main metabolites, with a precise and sensitive technique (liquid 14 chromatography-tandem mass spectrometry (LC-MS/MS), recruited for 13 months from January 2016 in the RCT and followed during 6 months postpartum.
With this more precise and sensitive technique, they show a decrease in the rate of vitamin D deficiency and insufficiency in both the mother and the offspring.

I have some minor comments

Since vitamin D levels depend on sun exposure, could the authors give Vitamin D rates for hours of walking away from home?

Could they stratify by body weight or categorized BMI?

Since the recruitment period lasted for 13 months, could you group vitamin D levels by quarter, to observe seasonal variability?

Could you estimate the cost of the technique?

Reviewer 2 Report

This paper reveals an interesting technique to detect a more appropriate number of (young) patients with vitamin D deficiency, making use of the detection of 3-epi-25(OH)D3. I have some comments:

  • What are the therapeutical consequences of these findings? Overall, there is some lack of evidence in the literature that vitamin D administration would prevent several diseases. The authors should address this in the Discussion.
  • I find the number of children with vitamin D deficiency to be quite high. Did the test subjects belong to some subpopulation?
  • Had cross-reactions with this epimer been ruled out?
